# Towards improving online learning in physical education: Gender differences and determinants of motivation, psychological needs satisfaction, and academic achievement in Saudi students

Mohamed Frikha[1,2]*, Nourhen Mezghanni[3☯], Nesrine Chaâri[2☯], Noureddine Ben Said[4], Mohammed S. Alibrahim[1], Majed M. Alhumaid[1], Marwa M. Hassan[1], Raghad S. Alharbi[1], Mostafa S. Amira[1], Nasser Abouzeid[1]

1 Department of Physical Education, College of Education, King Faisal University, Al-Ahsa, Saudi Arabia, 2 Research Laboratory Education, Motricity, Sport and Health (LR19JS01), High Institute of Sport and Physical Education, Sfax University, Sfax, Tunisia, 3 Department of Sport Sciences, College of Education, Taif University, Taif, Saudi Arabia, 4 Department of Biomechanics and Motor Behavior, College of Sport Sciences and Physical Activity, King Saud University, Riyadh, Saudi Arabia

☯ These authors contributed equally to this work.
* mfrikha@kfu.edu.sa, medfrimed@gmail.com

**Data Availability Statement:** data are held in a public repository Harvard Dataverse, DRAFT VERSION https://doi.org/10.7910/DVN/WQWLMU.

## Abstract

Studies on students' perceptions and expectations during physical education (PE) online learning remain scarce. Centered on self-determination theory, the present cross-sectional study aims to identify gender differences and predictors affecting motivation, psychological needs satisfaction (PNS), and academic achievement during PE online learning. Data were collected from Saudi students' (N = 308, 161 females and 147 males) responses to the PE autonomy, relatedness, competence, and motivation questionnaires. Welch's t-test for unequal sample sizes, multiple linear regression, and binary logistic regression were used to compare means and to predict the relationships between the independent and dependent variables. The results showed higher autonomy and competence perceptions in female than in male students, but no differences were observed in relatedness. Female students presented higher intrinsic motivations, lower amotivation perceptions than males. However, no gender differences were recorded in extrinsic motivation. Students with less experience in online learning and weak grade point averages (GPAs) are more susceptible to having a high level of amotivation. Gender, GPA, and prior experience with online learning are the common predictors for all PNS and amotivation, while GPA and prior experience with online learning are the determinants of intrinsic motivation. GPA is affected by prior experience with online learning, autonomy, competence, intrinsic motivation, and amotivation. Therefore, teachers are encouraged to adapt their didactic-pedagogical behaviors during PE online learning according to students' motivation and autonomy perceptions. Structuring teaching activities with more individualized support for autonomy, competence, intrinsic motivation, and students' online skills/competencies ensures better learning efficiency and academic achievements.

**Funding:** The corresponding author Frikha Mohamed received a grant for the present study. GRANT 3961 Funder: The Deanship of Scientific Research, King Faisal University, Al-Ahsa 31982, Saudi Arabia https://www.kfu.edu.sa/en/Deans/Research/Pages/Home-new.aspx The funders had no role in study design, data collection and analysis, decision to publish, or preparation of the manuscript.

## Introduction

Physical education (PE) and sports are essential to change human behavior and to improve self-esteem, social cohesion, and solidarity [1, 2]. Nonetheless, the swapped teaching methods from traditional, face-to-face, to online learning modalities (carried out synchronously or asynchronously), in relation to the Covid-19 pandemic restrictions and social distancing [3], may have induced changes in students' perceptions and expectations toward the learning process. Indeed, this learning modality requires students and teachers to familiarize themselves with new circumstances, to be able to use technology, to develop and maintain healthy constructive social interactions [4]. Despite the recent large number of studies conducted to assess the pandemic effects on education [5–8], research focusing on PE online learning remains scarce [9–11]. Furthermore, significant concerns were raised, particularly regarding online learning efficiency and student expectations. As previously mentioned, the use of online learning in PE generates a number of challenges, largely related to the weak effectiveness in motor skill acquisition and physical activity improvement in students [12], the lack of interactions (with teachers and peers), the disappearance of the sense of belonging to the institution, and the lack of common goals [13]. Therefore, investigating students' motivation and psychological needs satisfaction (PNS, i.e., autonomy, competence, relatedness) during PE online learning remains of great importance, as it allows for better apprehending this process and providing some possible solutions to the aforementioned issues.

### Psychological needs satisfaction and motivation

According to self-determination theory (SDT) [14], there are three universal basic needs: autonomy (feelings of volition, self-governance, self-acceptation or self-determination), competence (feelings of talent, ability, efficacy, and capability), and relatedness (feelings of belonging, intimacy, and connection to others). Consequently, human behavior can be intrinsically motivated (performing tasks without any external rewards), extrinsically motivated (external, introjected, identified, and integrated regulation), or amotivated (no intentions or tendency to engage in a behavior) [15]. Research evidence suggests that motivation should be taken seriously into account in the online learning environment [16]. Moreover, the results on gender differences in motivation and PNS are controversial [17]. Indeed, if some studies showed more controlled regulation in women (introjective) than in men [18], others affirmed the opposite: women's behavior seems to be more autonomous, and men's behavior is more externally regulated [19]. An earlier meta-analysis [20] demonstrated negligible gender differences in motivational regulation. While numerous studies supported the idea [21, 22], others demonstrated contrasting findings and identified no gender differences regarding motivational regulations [23]. Nevertheless, research suggesting the existence of differences between males' and females' motivation during the learning process remains unable to explain the origin of these differences [20].

### The Saudi context

The culture of sports practice in Saudi society is still absent, in particular for Saudi women [1]. Sports activities and PE sessions are newly introduced in school programs for women [2]. Moreover, Saudi society is generally conservative [24] and opportunities for social contact differ from men to women. Indeed, gender mixing (*ikhtilat)* [25] is still forbidden or limited to the close family environment and is strictly not allowed, even in schools and universities. This context exacerbates the gap between male and female students and maintains gender segregation, which represents a significant characteristic of public and social life in Saudi Arabia [24].

The scientific literature shows a gender gap in physical activity (PA) practice and demonstrates differences between males and females [26], with women around the world less likely than men to meet the minimum PA recommendations and then to have an active lifestyle. In concordance with, studies in the Saudi Arabia context reported similar barriers to women's participation in PA due to the lack of time, the lack of motivation [27], the scarcity of family support, and the high cost of PA practice [28]. These results suggest that there is reason to expect differences and possible relationships between PNS and motivational regulations in male and female PE students in the Saudi context during online learning.

## The present study

Recently, it was demonstrated that Saudi female PE students presented higher psychological needs satisfaction (PNS) than males (ie. autonomy, competence, and relatedness) during online learning sessions [10]. The PNSs were found to be related to the type of physical activities undertaken, to the daily sleep hours, and to the previous experiences with the online learning modality. In a review, Murtagh et al. [29] concluded that online teaching and learning in PE could have an impact on the establishment of relationships, either positively or negatively. Previous studies [30, 31] highlighted that female integration in physical activities faces numerous problems. The most relevant are related to the lack of motivation, the lack of awareness among female students, the lack of understanding of its importance, and the loss of interest. Nevertheless, it was reported that academic integration has a mediating effect on the relationship between PNS and students' intrinsic motivation [32, 33]. Thus, academic integration as a psychological process fosters students' intellectual development [34] and promotes the enhancement of intrinsic motivation, one of the most influential factors in students' academic achievements [34, 35].

To the author's best knowledge, few studies have focused on psychological processes and the exploration of gender-related differences in motivations and PNS during online learning [10, 31]. Therefore, the present investigation aims to first identify the gender differences in PNS, motivations, during the online learning. Second, to explore the relationships and the determinants affecting students' PNS, motivation, and academic achievement during PE online sessions. Hence, to achieve these goals, many questions arise in relation to the differences in perception between males and females in PNS and motivation during online learning in physical education. Which of the predictors (gender, prior experience with online learning, and GPA) can be in relation to PNS and motivation during online learning in physical education? Which of the predictors (gender, prior experience with online learning, motivation, and PNS) affect the student's academic achievement during online learning in physical education?

According to the aforementioned question, we advanced the following study hypothesis:

**H1.** Students' PNS and motivation perceptions during online learning in physical education vary significantly according to the variables gender, GPA, and prior experience with online learning.

**H2.** The variables gender, GPA, and prior experience with online learning, significantly predict students' PNS and motivation during online learning in physical education.

**H3.** Students' GPAs are significantly determined by their prior experience with online learning, the levels of autonomy, competence, intrinsic motivation, and amotivation perceptions.

## Materials and methods

### Study design and sample size

The present study follows a quantitative cross-sectional design. According to the geographical regions in Saudi Arabia, we randomly selected three from the seven public universities in which PE or sports science programs are conducted. In total, 705 students (382 males and 323 females) were active, as declared by the dean of Information Technology in each university. The minimum required sample size, (n), was calculated according to Daniel [36]:

$n = [z^2 \times p \times (1 - p) / e^2] / [1 + (z^2 \times p \times (1 - p) / (e^2 \times N)]$; where N = population size; p = sample proportion (50%); confidence level (95%); z score (1.96); and e = margin of error (5%).

Taking into account the possible dropout rate of 20% [10], the total required sample size was set at 299 PE students. Nonetheless, to ensure better student contributions, the questionnaire link was emailed to almost all active PE students in the selected universities. In total, 705 requests were sent. The final sample size of the study was set at 308 (147 males and 161 females), which indicated that the study sample size was appropriate. The King Faisal University Ethics Committee approved this study (KFU-REC-2020- JAN -ETHICS440).

### Study population

Physical education students from King Faisal University, King Saud University, and Taif University were invited to participate in the study from November 18 to December 18, 2020. The teaching process during this period and since March 8, 2020, was conducted online in all Saudi universities in response to the COVID-19 pandemic restrictions [3]. Thus, all courses and examinations were performed exclusively via the Blackboard® platform. Hence, all participants received online learning in PE courses (i.e., practical and theoretical subjects) for at least seven months. The inclusion criteria were as follows: (i) being a PE student; (ii) being a regular student: students with attendance < 75% were not invited to participate; and (iii) responding to all questions of the questionnaire. All participants were assured about the anonymity of their responses and no information that may reveal their identity was required. All respondents, who volunteered, were informed about the objective of the study, approved written informed consent, and agreed to have their responses published by checking a specific box at the beginning of the questionnaire indicating their acceptance to participate.

### Questionnaires

In the present investigation, we used items from two questionnaires: the PE autonomy relatedness competence (PE-ARCS) and the PE motivation (PE-MS) scales of Sulz et al. [37]. Therefore, the present study questionnaire was composed of three parts. The first part collected demographic data (age, gender, weight, height), grade point average (GPA; $3 < GPA \leq 5$ and $1 < GPA \leq 3$), and experience in online learning. The second part of the questionnaire involved 12 items from the PE-ARCS for which students were asked to estimate their self-perceptions during the online physical education classes. Four items were assigned to each subsection: autonomy (items 3, 6, 9, and 12), competence (items 2, 5, 8, and 11), and relationship (items 1, 4, 7, and 10). In the third part, the nine PE-MS items were provided. Participants were asked to rate their perceptions regarding intrinsic motivation (items 13, 16, 19), extrinsic motivation (items 14, 17, 20), and amotivation (items 15, 18, 21). Responses were given on a 5-point Likert scale (1 = strongly disagree, 2 = disagree, 3 = neutral, 4 = agree, 5 = strongly agree).

To ensure a better understanding of the questionnaire items, the English versions of the PE-ARCS and PE-MS were translated into Arabic and adapted to the Saudi context by two

bilingual translators, following the recommendations of Beaton et al. [38]. The translation was based essentially on the meaning of the English statements and not only on the textual rendering. Then, two English speakers of Arab origin back-translated this version into English, and three bilingual experts reviewed both versions (Arabic and English) and made the necessary adjustments. No items were removed during the translation process. The obtained Arabic versions of the PE-ARCS and the PE-MS (S1 and S2 Files) were tested for validity and reliability through a pilot study conducted with a sample of PE students (N = 31) out of the study sample and using exploratory factor analysis and Cronbach's alpha for factorial validity and internal reliability, respectively. All Cronbach's alpha values were satisfactory, with 0.758 for autonomy, 0.811 for competence, 0.761 for relatedness, 0.823 for intrinsic motivation, 0.836 for extrinsic motivation, and 0.849 for amotivation.

## Data analysis

The statistical analysis was performed using SPSS V.26 (IBM, Armonk, NY, USA). Descriptive data were analyzed by calculating the mean, standard deviation (M ± SD), or proportions of the total population. Normality was tested using histograms and absolute values of skewness and kurtosis, and all values were < 2 [39]. Exploratory factor analysis (EFA) using principal component analysis and Cronbach's alpha was used to verify the validity and reliability of the questionnaires. For mean comparisons between the groups, stratified by gender, prior experience with online learning, and GPAs, we used Welch's t-tests for unequal sample sizes (equal variance not assumed). Multiple linear regression (MLR) for categorical variables was performed to predict the relationship between the categorical variables (gender, experience with online learning, and GPA) and the dependent variables (autonomy, competence, relatedness, intrinsic motivation, extrinsic motivation, and amotivation). However, Binary logistic regression was performed to predict factors that may affect GPA (high GPA $3 < GPA \leq 5$; and low GPA $1 < GPA \leq 3$). The effect size (eta partial squared $\eta^2$) was calculated and classified as small (0.1), medium (0.3), and large ($> 0.5$) [40]. The significance level was set at $p < 0.05$.

## Results

### Validity and reliability of the Arabic versions of the PE-ARCS and PE-MS

Exploratory factor analysis (EFA) was used using the principal component analysis extraction approach and Varimax rotation. For the PE-ARCS, the Kaiser–Meyer–Olkin (KMO) sampling adequacy was above the acceptable threshold of 0.6 (KMO = 0.658); Bartlett's sphericity test result was statistically significant ($p < 0.001$). Three components with eigenvalues $> 1$ (5.298, 1.929, and 1.315) were identified and maintained (cumulative variance was set at 71.182%). For the PE-MS, the Kaiser–Meyer–Olkin (KMO) sampling adequacy was above the acceptable threshold of 0.6 (KMO = 0.671); Bartlett's sphericity test result was statistically significant ($p < 0.001$). Three components with eigenvalues $> 1$ (3.654, 2.066, and 1.137) were identified and maintained (cumulative variance was set at 76.192%).

Concerning the internal reliability, Cronbach's alpha values were set at 0.869 for PE-ARCS (all items included) and 0.705, 0.730, and 0.682 for autonomy, relatedness, and competence, respectively. For the PE-MS, Cronbach's alpha values were set at 0.851 (all items included) and 0.830, 0.799, and 0.855 for intrinsic motivation, extrinsic motivation, and amotivation, respectively. Therefore, the Arabic versions of the PE-ARCS and the PE-MS scales were found to be both reliable and valid for measuring PNS and motivations, respectively.

**Table 1. Sociodemographic characteristics of the respondents (*n* = 308).**

| Variables | Values * |
|---|---|
| **Gender** | |
| Male | 147 (47.7%) |
| Female | 161 (52.3%) |
| **Age** | 21.92 ± 1.35 years |
| **Body mass index (BMI)** | |
| Male | 24.43 ± 3.61 kg/m² |
| Female | 20.19 ± 2.80 kg/m² |
| **Universities** | |
| King Faisal University (KFU) | 79 (25.6%) |
| King Saud University (KSU) | 75 (24.4%) |
| Taif University (TU) | 154 (50.0%) |
| **Prior experience with online learning** | |
| Yes | 152 (49.4%) |
| No | 156 (50.6%) |
| **Grade point average (GPA)** | |
| $3 < GPA \leq 5$ | 197 (44.7%) |
| $1 < GPA \leq 3$ | 111 (25.2%) |

* Values are given as *n* (%) unless otherwise stated.

## Sociodemographic characteristics

The sample size was set at 308 PE students (147 males and 161 females), representing a response rate of 43.7%. Descriptive statistics of the variables gender, age, body mass index, universities, previous experiences with online learning, and GPA are presented in Table 1.

## Comparisons in PNS and motivation according to the gender variable

Concerning the variable gender, Welch's t-test (equal variance not assumed) revealed higher values in female compared to male students on competence and autonomy ($p = 0.001$, $\eta^2 = 0.138$, small; $p = 0.001$, $\eta^2 = 0.255$, medium, respectively). However, no significant difference was detected in relatedness ($p = 0.086$).

Concerning motivation, the statistical analysis showed higher values in intrinsic motivation ($p = 0.002$; $\eta^2 = 0.115$, small) and extrinsic motivation ($p = 0.127$; $\eta^2 = 0.055$, small) in female students than in male students. However, for amotivation, lower values were recorded in females ($p = 0.001$; $\eta^2 = 0.377$, medium) (Table 2).

## Comparisons in PNS and motivation according to GPA

According to GPA, the results showed higher values in autonomy ($p = 0.001$, $\eta^2 = 0.267$, medium), competence ($p = 0.001$, $\eta^2 = 0.321$, medium), relatedness ($p = 0.001$, $\eta^2 = 0.223$, medium), and intrinsic motivation ($p = 0.001$, $\eta^2 = 0.255$, medium) in group with high GPA: (I) ($3 < GPA \leq 5$) than in group with low GPA: (II) ($1 < GPA \leq 3$). However, no significant difference was detected between the two groups in extrinsic motivation ($p = 0.088$). Moreover, concerning amotivation, lower values were observed in group I than in group II ($p = 0.001$, $\eta^2 = 0.216$, medium) (Table 3).

Comparisons between group I ($3 < GPA \leq 5$) and group II ($1 < GPA \leq 3$) in males only showed higher autonomy, competence, and relatedness in group I ($p = 0.001$, $\eta^2 = 323$,

**Table 2. Effect of the variable gender on psychological needs satisfaction and motivation (M ± SD).**

|  | Gender | N | Mean ± SD | MD (SE) | Welch's t test† | Sig. | 95% CI of MD (L–U) |
|---|---|---|---|---|---|---|---|
| **Autonomy** | male | 147 | 15.03 ± 2.907 | - 1.873 (0.295) | 6.345 | 0.001*** | - 2.455 –-1.292 |
|  | female | 161 | 16.90 ± 2.186 |  |  |  |  |
| **Competence** | male | 147 | 15.35 ± 3.408 | - 1.894 (0.338) | 5.594 | 0.001*** | - 2.562–1.228 |
|  | female | 161 | 17.25 ± 2.398 |  |  |  |  |
| **Relatedness** | male | 147 | 14.51 ± 2.964 | - 5.395 (0.313) | 1. 723 | 0.086 | - 1.156–0.077 |
|  | female | 161 | 15.05 ± 2.482 |  |  |  |  |
| **Intrinsic motivation** | male | 147 | 10.96 ± 2.657 | - 0.897 (0.292) | 3.075 | 0.002** | - 1.471 –- 0.323 |
|  | female | 161 | 11.86 ± 2.445 |  |  |  |  |
| **Extrinsic motivation** | male | 147 | 13.08 ± 2.401 | 0.436 (0.285) | 1.528 | 0.127 | - 1.125–0.997 |
|  | female | 161 | 12.65 ± 2.608 |  |  |  |  |
| **Amotivation** | male | 147 | 10.01 ± 2.474 | 1.901 (0.353) | 5.391 | 0.001*** | 1.206–2.594 |
|  | female | 161 | 8.09 ± 3.647 |  |  |  |  |

*Significantly different at: * $p < 0.05$

*** $p < 0.001$

† Equal variance not assumed; CI confidence interval; MD mean difference; SE standard error

moderate; $p = 0.001$, $\eta^2 = 333$, moderate; and $p = 0.001$, $\eta^2 = 0.193$, small, respectively). Higher intrinsic motivation and lower amotivation in group I ($p = 0.004$; $\eta^2 = 0.239$, medium; $p = 0.003$, $\eta^2 = 0.186$, small, respectively) compared to group II. However, no differences were recorded in extrinsic motivation between the two groups ($p = 0.852$).

Concerning the female results, higher values were recorded in group I than in group II in autonomy ($p = 0.001$, $\eta^2 = 0.178$, small), in competence ($p = 0.001$, $\eta^2 = 0.314$, medium), in relatedness ($p = 0.002$, $\eta^2 = 0.244$, medium), in intrinsic motivation ($p = 0.001$, $\eta^2 = 242$, medium), in extrinsic motivation ($p = 0.001$, $\eta^2 = 197$, small). However, lower amotivation values were recorded in group I compared to group II ($p = 0.001$, $\eta^2 = 0.243$, medium).

**Table 3. Psychological needs satisfaction and motivation values according to GPA (M ± SD).**

|  | GPA | N | Mean ± SD | MD (SE) | Welch's t test† | Sig. | 95% CI of MD (L–U) |
|---|---|---|---|---|---|---|---|
| **Autonomy** | (I) | 197 | 16.99 ± 2.047 | 2.743 (0.309) | 8.850 | 0.001*** | 2.131–3.354 |
|  | (II) | 111 | 14.25 ± 2.881 |  |  |  |  |
| **Competence** | (I) | 197 | 17.58 ± 2.143 | 3.425 (0.344) | 9.964 | 0.001*** | 2.747–4.104 |
|  | (II) | 111 | 14.15 ± 3.245 |  |  |  |  |
| **Relatedness** | (I) | 197 | 15.69 ± 2.277 | 2.492 (0.307) | 8.103 | 0.001*** | 1.886–3.099 |
|  | (II) | 111 | 13.20 ± 2.753 |  |  |  |  |
| **Intrinsic motivation** | (I) | 197 | 11.07 ± 2.585 | 2.371 (0.297) | 8.120 | 0.001*** | 1.794–2.946 |
|  | (II) | 111 | 10.33 ± 2.668 |  |  |  |  |
| **Extrinsic motivation** | (I) | 197 | 13.04 ± 2.479 | 0.514 (0.301) | 1.711 | 0.088 | - 0.078–1.106 |
|  | (II) | 111 | 12.53 ± 2.561 |  |  |  |  |
| **Amotivation** | (I) | 197 | 7.91 ± 3.186 | - 3.032 (0.324) | -9.340 | 0.001*** | - 3.671 –- 2.393 |
|  | (II) | 111 | 10.94 ± 2.445 |  |  |  |  |

GPA: grade point average; (I) high GPA group: $3 < GPA \leq 5$; (II) low GPA group: $1 < GPA \leq 3$

*Significantly different from group (II) at ** $p < 0.01$

*** $p < 0.001$

† Equal variance not assumed; CI confidence interval; MD mean difference; SE standard error

**Table 4. Psychological needs satisfaction and motivation values according to the prior experience to online learning (M ± SD).**

|  | Prior experience to OL | N | Mean ± SD | MD (SE) | Welch's t test† | Sig. | 95% CI of MD (L–U) |
|---|---|---|---|---|---|---|---|
| **Autonomy** | yes | 152 | 17.65 ± 1.7 | 3.247 (0.247) | 13.129 | 0.001*** | 2.760–3.734 |
|  | no | 156 | 14.4 ± 2.565 |  |  |  |  |
| **Competence** | yes | 152 | 18.23 ± 1.701 | 3.724 (0.276) | 13.473 | 0.001*** | 3.179–4.268 |
|  | no | 156 | 14.51 ± 2.991 |  |  |  |  |
| **Relatedness** | yes | 152 | 16.39 ± 2.138 | 3.151 (0.254) | 12.393 | 0.001*** | 2.651–3.651 |
|  | no | 156 | 13.24 ± 2.323 |  |  |  |  |
| **Intrinsic motivation** | yes | 152 | 13.00 ± 1.718 | 3.089 (0.235) | 13.132 | 0.001*** | 0.472–1.679 |
|  | no | 156 | 9.91 ± 2.367 |  |  |  |  |
| **Extrinsic motivation** | yes | 152 | 13.49 ± 1.812 | 1.249 (0.276) | 4.517 | 0.001*** | - 0.470–0.462 |
|  | no | 156 | 12.24 ± 2.927 |  |  |  |  |
| **Amotivation** | yes | 152 | 6.89 ± 2.589 | - 4.169 (0.289) | -14.440 | 0.001*** | - 1.713 –-0.291 |
|  | no | 156 | 11.06 ± 2.474 |  |  |  |  |

OL: online learning

*Significantly different from opposite response at: * $p < 0.05$; ** $p < 0.01$; *** $p < 0.001$

† Equal variance not assumed; CI confidence interval; MD mean difference; SE standard error

## Comparisons in PNS and motivation according to prior experience with online learning variable

Concerning PNS, Welch's t-test showed higher perceptions of autonomy, competence, and relatedness in the more experienced group with online learning ($p = 0.001$, $\eta^2 = 0.416$, medium; $p = 0.001$, $\eta^2 = 0.433$, medium; $p = 0.001$, $\eta^2 = 0.425$, medium, respectively). Concerning intrinsic and extrinsic motivations, higher values were observed in the group declaring to have previous experiences in online learning ($p = 0.001$, $\eta^2 = 0.382$, medium; $p = 0.002$, $\eta^2 = 0.099$, small). In contrast, the same group presented lower amotivation scores compared to the group having no previous experiences in online learning ($p = 0.001$, $\eta^2 = 0.428$, moderate) (Table 4).

## Multiple linear regression: Predictors of PNS and motivation

Table 5 illustrates the importance, directions, and strength of relationships between the predictors (gender, GPA, and prior experience with online learning) and the dependent variables (autonomy, competence, relatedness, intrinsic motivation, extrinsic motivation, and amotivation).

The MLR showed that gender, prior experience with online learning and GPA are the common predictors affecting positively autonomy ($R^2 = 0.442$; $\beta = 0.887$, $p < 0.001$; $\beta = 2.392$, $p < 0.001$, and $\beta = 1.425$, $p < 0.001$, respectively), competence ($R^2 = 0.470$; $\beta = 0.698$, $p < 0.01$; $\beta = 2.665$, $p < 0.001$, and $\beta = 2.028$, $p < 0.001$, respectively), relatedness ($R^2 = 0.384$; $\beta = 0.524$, $p < 0.05$; $\beta = 2.217$, $p < 0.001$, and $\beta = 1.363$, $p < 0.001$, respectively), but negatively affecting amotivation ($R^2 = 0.447$; $\beta = 0.651$, $p < 0.05$; $\beta = -3.439$, $p < 0.001$, and $\beta = -1.289$, $p < 0.001$, respectively).

Intrinsic motivation was affected ($R^2 = 0.397$) by prior experience with online learning and GPA ($\beta = 2.611$, $p < 0.05$; and $\beta = 1.188$, $p < 0.001$, respectively). Extrinsic motivation was weakly affected ($R^2 = 0.088$) by gender and experience with online learning ($\beta = 0.858$, $p < 0.01$; and $\beta = 1.473$, $p < 0.001$, respectively).

**Table 5. Multiple linear regression between the categorical variables (gender, prior experience with online learning, and GPA), PNS and motivation scores.**

| DV | | β (SE) | R² | t | Sig. | 95% CI for β |
|---|---|---|---|---|---|---|
| | | | | | | L–U |
| Auto | Constant (N = 308) | 14.337 (0.259) | 0.442 | 55.283 | <0.001*** | 13.827–14.848 |
| | Gender (1 = male; 0 = female) | - 0.887 (0.244) | | - 3.631 | <0.001*** | - 1.367 – - 0.406 |
| | Experience with OL (1 = yes; 0 = no) | 2.392 (0.265) | | 9.031 | <0.001*** | 1.871–2.914 |
| | GPA (1: 3 < GPA ≤ 5; 0: 1 < GPA ≤ 3 | 1.425 (0.272) | | 5.238 | <0.001*** | 0.890–1.961 |
| Comp | Constant (N = 308) | 14.065 (0.286) | 0.470 | 49.264 | <0.001*** | 13.503–14.627 |
| | Gender (1 = male; 0 = female) | - 0.698 (0.269) | | - 2.597 | 0.01** | - 1.227 – - 0.169 |
| | Experience with OL (1 = yes; 0 = no) | 2.665 (0.292) | | 9.139 | <0.001*** | 2.091–2.617 |
| | GPA (1: 3 < GPA ≤ 5; 0: 1 < GPA ≤ 3 | 2.028 (0.300) | | 6.768 | <0.001*** | 1.438–2.617 |
| Relat | Constant (N = 308) | 12.329 (0.274) | 0.384 | 44.989 | <0.001*** | 11.790–12.868 |
| | Gender (1 = male; 0 = female) | 0.524 (0.258) | | 2.032 | 0.043* | 0.017–1.032 |
| | Experience with OL (1 = yes; 0 = no) | 2.717 (0.280) | | 9.707 | <0.001*** | 2.166–3.268 |
| | GPA (1: 3 < GPA ≤ 5; 0: 1 < GPA ≤ 3 | 1.363 (0.288 | | 4.741 | <0.001*** | 0.797–1.929 |
| IM | Constant (N = 308) | 9.340 (0256) | 0.397 | 36.437 | <0.001*** | 8.835–9.844 |
| | Experience with OL (1 = yes; 0 = no) | 2.611 (0.262) | | 9.973 | <0.001*** | - 0.377–0.573 |
| | GPA (1: 3 < GPA ≤ 5; 0: 1 < GPA ≤ 3 | 1.188 (0.269 | | 4.418 | <0.001*** | 0.659–1.718 |
| EM | Constant (N = 308) | 11.698 (0.307) | 0.088 | 38.096 | <0.001*** | 11.094–12.303 |
| | Gender (1 = male; 0 = female) | 0.858 (0.289) | | 2.967 | 0.003** | 0.289–1.427 |
| | Experience with OL (1 = yes; 0 = no) | 1.473 (0.314) | | 4.696 | <0.001*** | - 0.594–0.674 |
| Am | Constant (N = 308) | 11.217 (0.311) | 0.447 | 36.016 | <0.001*** | 10.604–11.830 |
| | Gender (1 = male; 0 = female) | 0.651 (0.293) | | 2.220 | 0.027* | 0.074–1.228 |
| | Experience with OL (1 = yes; 0 = no) | - 3.439 (0.318) | | - 10.809 | <0.001*** | - 4.065 – - 2.813 |
| | GPA (1: 3 < GPA ≤ 5; 0: 1 < GPA ≤ 3 | - 1.289 (0.327) | | - 3.943 | <0.001*** | - 1.932 – - 0.645 |

Auto: autonomy; Comp: competence; Relat: relatedness; IM: intrinsic motivation; EM: extrinsic motivation; Am: amotivation; OL: online learning; GPA: grade point average; β: unstandardized coefficient; SE: standard errors; DV: dependent variable

*Significantly different at: * p <0.05; ** p <0.01; *** p <0.001

## Binary logistic regression: Predictors of GPA

Considering GPA as a dependent variable (high GPA group 3 < GPA ≤ 5; and low GPA group 1 < GPA ≤ 3); and gender, prior experience with online learning, PNS, and motivation as predictors, the Binary logistic regression showed the following (Table 6). First, the model

**Table 6. Binary logistic regression results for assessing the impact of the predictors on GPA.**

| DV | | β (SE) | Sig. | Nagelkerke R² | Exp (β) | 95% CI for Exp(β) |
|---|---|---|---|---|---|---|
| | | | | | | L–U |
| GPA (0: 1 < GPA ≤ 3; 1: 3 < GPA ≤ 5) | Constant (N = 308) | - 11.910 (2.689) | <0.001*** | 0.553 | 0.000 | |
| | Prior experience with OL (1 = yes; 0 = no) | 1.851 (0.577) | 0.001*** | | 6.364 > 1 | 2.054–19.714 |
| | Autonomy | 0.406 (0.095) | <0.001*** | | 1.501 > 1 | 1.246–1.808 |
| | Competence | 0.292 (0.085) | 0.001*** | | 1.339 > 1 | 1.135–1.580 |
| | IM | 0.281 (0.079) | <0.001*** | | 1.324 > 1 | 1.133–1.546 |
| | AM | - 0.314 (0.074) | <0.001*** | | 0.731 < 1 | 0.632–0.845 |

GPA: grade point average; OL: online learning; β: unstandardized coefficient; SE: standard errors; DV: Dependent variable

***Significantly different at p <0.001

showed a good fit, and described the data very well, as the Omnibus test of the model coefficient was very significant (p < 0.001). Second, the Hosmer and Lemeshow test showed no significance (p = 0.432) and almost equal values between the observed and expected values. Hence, there are no differences between the observed and predicted models.

Nagelkerke ($R^2$ = 0.553) showed that 55.3% of the changes in the criterion variables can be accounted for by the predictor variables in the model. Considering the dependent variable (GPA) encoding (0: 1 < GPA ≤ 3; and 1: 3 < GPA ≤ 5), positive and significant relationships were observed between GPA and prior experience with online learning, autonomy, competence, and intrinsic motivation. However, a significant negative relationship is observed with AM, indicating that the higher the GPA is, the lower the AM score. Moreover, the high value of the odds ratio (Exp (β) = 6.364; > 1) for the variable prior experience with online learning, shows that the probability of falling into group 1 (3 < GPA ≤ 5) is greater than the probability of falling into group 0 (1 < GPA ≤ 3). Hence, we can say that the odds of students having a high GPA (3 < GPA ≤ 5) and high prior experience with online learning are 6.364 times higher than those of the group having low GPA (1 < GPA ≤ 3) and low prior experience with online learning (Table 6).

Concerning autonomy, the high value of the odds ratio (Exp (β) = 1.501; > 1) indicates that the probability of falling into group 1 (3 < GPA ≤ 5) is greater than the probability of falling into group 0 (1 < GPA ≤ 3). Hence, we can say that the odds of students having a high GPA (3 < GPA ≤ 5) and high autonomy perception are 1.501 times higher than those of the group having low GPA (1 < GPA ≤ 3) and low autonomy perception.

Concerning competence, the high value of the odds ratio (Exp (β) = 1.339; > 1) indicates that the probability of falling into group 1 (3 < GPA ≤ 5) is greater than the probability of falling into group 0 (1 < GPA ≤ 3). Hence, we can say that the odds of students having a high GPA (3 < GPA ≤ 5) and high competence perception are 1.339 times higher than those of the group having low GPA (1 < GPA ≤ 3) and low competence perception.

Concerning intrinsic motivation, the high value of the odds ratio (Exp (β) = 1.324; > 1) indicates that the probability of falling into group 1 (3 < GPA ≤ 5) is greater than the probability of falling into group 0 (1 < GPA ≤ 3). Hence, we can say that the odds of students having a high GPA (3 < GPA ≤ 5) and high intrinsic motivation are 1.324 times higher than those of the group having low GPA (1 < GPA ≤ 3) and low intrinsic motivation (Table 6).

## Discussion

The present investigation aimed to identify the gender differences in PNS, motivation, and academic achievements, as well as to explore the relationships and factors that affect it during PE online learning sessions. The main findings were as follows: (i) Higher autonomy and competence values were recorded in female than in male students. Female students presented higher intrinsic motivation but lower amotivation perceptions compared to males. (ii) Students with less prior experience with online learning and weak GPAs are more susceptible to having a high level of amotivation. (iii) Gender, GPA, and prior experience with online learning are the common predictors for all PNS and amotivation, while GPA is affected by prior experience with online learning, autonomy, competence, intrinsic motivation, and amotivation perceptions.

Although there are several related studies [23, 32] exploring motivation, psychological needs satisfaction, and academic achievement in a habitual learning context, the investigation of factors that can affect it during the online learning context remains relevant [10] to the extent that it can offer new perspectives in the implementation of personalized learning approaches focusing on learners' characteristics and preferences [41, 42].

Saudi women's participation in physical education is relatively new, as is the unfamiliar teaching process (the suspended attendance in schools and universities due to the COVID-19 pandemic lockdown, which lacks direct communication with teachers and peers); they reported higher perception of some PNSs (i.e., autonomy and competence), intrinsic motivations, and lower amotivation compared to men. The present findings are in accordance with previous studies showing higher autonomy, competence, and relatedness perceptions in female than in male PE students [10]. Other studies [18, 19] reported that women are more autonomous and men are more externally regulated in relation to exercise behavior. Thus, women generally tend towards more controlled regulations than men do.

In general, scientific literature findings related to men's and women's differences with respect to motivation for exercise are still controversial [20]. Indeed, while many studies reported the existence of gender differences [21, 22], others found negligible gender differences [20] or revealed no significant gender differences in motivational regulations [23]. The present study found higher female perceptions of intrinsic and extrinsic motivation than male perceptions, which is in accordance with previous studies [21, 22]. Nonetheless, it is still at odds with recent findings, revealing that adolescent males showed a greater presence of motivation towards engaging in PA, while females showed more barriers [43]. The discrepancies can be explained by the difference in sample characteristics (i.e., age, culture) and/or the differences in measurement tools of motivation. Research suggesting the existence of differences between males' and females' motivation during the learning process remains unable to explain the origin of these differences. Nonetheless, discrepancies could be related to some biological/genetic, environmental, and social reasons [44], to the sample characteristics used in different studies [22], to the psychometric consistency of the measurement tools used [20], to the personal characteristics of the teacher and the adopted teaching style [11, 22] or to the type of feedback administered [45]. Indeed, teachers' differential behaviors toward males and females in class sessions and the nature of feedback used during the teaching-learning process [22, 45, 46] could represent a serious argumentation of male-female motivation differences. In this context, it was recently confirmed that "*directive feedback had a negative correlation with male students' intrinsic motivation and a positive correlation with female students' extrinsic motivation*" [45].

As reported by Aljehani et al. [2], two main factors may explain Saudi females' engagement with physical activity: the first is personal motivation (intrinsic) produced from an internal desire to be physically active and have a healthy lifestyle, and the second is family support (extrinsic motivation), which includes encouragement and praise. According to the socioecological model of Elder et al. [47], in addition to the intrapersonal and interpersonal factors, others may influence female behaviors such as the community, social institutions, and public policy (national, state, local laws, and regulations) [48]. Therefore, the high intrinsic and extrinsic motivation in Saudi females could be explained by their high desire for self-affirmation to explore a new domain, as well as by the continuous state attention aiming to meet the objectives of the Saudi 2030 Vision.

A recent study conducted by AL-Shahrani [1] demonstrated several motives that are the origin of Saudi women's sports practice: achieving fitness, agility, health preservation, and self-satisfaction. Therefore, the higher intrinsic and extrinsic motivations of females can be explained by their satisfaction and increased degree of self-determination. In contrast, the higher amotivation values in male students compared to females indicate their low satisfaction and enjoyment perceptions during the online PE sessions. It seems that the conversion from the face-to-face to the online learning modality affected their intrinsic motivation and then their desire to assist in the online PE sessions.

Students with higher GPAs and prior experiences with online learning demonstrated better PNS and intrinsic motivation, which supports the idea of their better academic achievement. The GPA, indicating better academic achievement of students, was demonstrated to be related to their academic integration [33, 49]. Indeed, academic integration as a psychological process supports the improvement of intrinsic motivation and is considered the most influential factor that assists students' intellectual development and their academic achievements [34, 35]. While academic integration was not directly measured in the present study, we can assume that the academic achievement of participants is essentially related to their intrinsic motivation. In line with our expectations and with the aforementioned studies, Vergara-Morales and Del Valle [32] concluded that academic integration partially mediated the relationship between students' PNS and intrinsic motivation.

The results of the MLR indicated that female students with more prior experience with online learning and a high GPA are likely to have high PNS, but low amotivation scores. Likewise, students with more prior experience in online learning and high GPAs are more likely to have high intrinsic motivation. However, the Binary logistic regression indicated that academic achievement is principally affected by autonomy, competence, intrinsic motivation, and prior experience with online learning. This finding seems to be in accordance with previous research suggesting improving students' autonomy in the classroom context. Indeed, Alrabai [50] and Reeve & Cheon [51] emphasized the need for autonomy-supportive activities in the classroom to enhance PNS. Others have mentioned the importance of competence-supportive activities in class sessions in the improvement of decision-making [52]. Likewise, the finding highlights the importance of improving students' competencies in online learning by strengthening their experiences and information technology skills [53], which was recommended in improving PNS, motivation, and academic achievement [54, 55].

In the PE context, Tilga et al. [11] reported that autonomy-supportive interventions for PE teachers should be directed toward combining face-to-face and online interventions to achieve the greatest effects on cognitive, organizational, and procedural behaviors, as well as on PNS and intrinsic motivation. The discussed findings emphasize the importance of autonomy, competence, and intrinsic motivation in high academic achievement. Thus, opting for its improvement in PE students requires developing a personalized approach that takes into account gender differences [56]. In this context, the individualized learning process was previously defined as an "*approach that provides learning choices and tailors learning content toward individuals' learning needs, interests, goals, and prior experiences to enhance knowledge and skills acquisition and support psychological need satisfaction and intrinsic motivation*" [42]. It was confirmed that individualized learning is an effective method for promoting students' engagement in classrooms and enhancing PNS and intrinsic motivation [42, 56].

## Strengths, limitations, and perspectives

The present study strived to clarify the differences and relationships between PNS, motivation, and academic achievement, as well as the associated contributing variables such as gender or experience with online learning in the PE context. Notwithstanding the findings of the present investigation, having a direct impact on the online PE teaching process, in structuring the teaching content (didactic transposition) [57], in making sense of the learner-knowledge relationship (relationship to knowledge), as well as in the learner-teacher relationship (didactic contract) [58], some limitations merit discussion. First, the cross-sectional design of the present study may not be suitable for proving causal relations between the study variables. Thus, longitudinal research designs are needed to better understand students' PNS and motivational profiles and their changes from face-to-face to online learning [10, 11]. Second, studying the

efficiency of online PE learning requires verifying the effect of new learning approaches. Issues based on the combination of both synchronous and asynchronous modalities [4] or the use of a flipped classroom model [59] could represent an interesting future research orientation. Moreover, the teachers' in-class behavior is considered a key factor that influences students' motivation [60]. Thus, promoting students' motivation during PE online learning according to an individualized specific classification system of teachers' motivational behaviors remains an interesting future research topic too [60].

## Conclusions

The present study demonstrates that during online learning in physical education, the Saudi female students presented higher autonomy, competence, and intrinsic motivations but lower amotivation perceptions than males. Students with less prior experience with online learning and weak GPAs are more susceptible to having a low level of PNS and intrinsic motivation, but a high level of amotivation. The predictors: gender, GPA, and prior experience with online learning positively affected all PNSs (autonomy, competence, and relatedness) and negatively affected the amotivation. However, the academic achievement was positively predicted by the variables: prior experience with online learning, autonomy, competence, and intrinsic motivation.

The basic PNS (i.e., autonomy, competence, and relatedness) and intrinsic motivation should not only be encouraged in students but should also be considered fundamental for structuring teaching content during online learning in PE. The gender differences emphasize the need for a personalized approach during PE online learning. Therefore, individualizing teachers' didactical/pedagogical behaviors through activities supporting autonomy, competence, intrinsic motivation, students' skills, and competencies in online PE learning ensures better learning efficiency and academic achievements.

## Supporting information

**S1 File. Physical Education Autonomy Relatedness Competence Scale (PEARCS Arabic version).**
(DOCX)

**S2 File. Physical Education Motivation Scale (PEMS Arabic version).**
(DOCX)

## Acknowledgments

The authors thank all of the subjects who participated in this study. Special thanks are given to the faculty members and IT staff administrations at KFU, KSU, and TU for their help.

## Author Contributions

**Conceptualization:** Mohamed Frikha, Nesrine Chaâri, Noureddine Ben Said, Mohammed S. Alibrahim, Majed M. Alhumaid.

**Data curation:** Nourhen Mezghanni, Nesrine Chaâri, Noureddine Ben Said, Majed M. Alhumaid, Marwa M. Hassan, Raghad S. Alharbi, Mostafa S. Amira, Nasser Abouzeid.

**Formal analysis:** Mohamed Frikha, Nourhen Mezghanni.

**Funding acquisition:** Mohamed Frikha, Majed M. Alhumaid.

**Investigation:** Nourhen Mezghanni, Nesrine Chaâri, Mohammed S. Alibrahim, Marwa M. Hassan, Raghad S. Alharbi, Mostafa S. Amira, Nasser Abouzeid.

**Methodology:** Mohamed Frikha, Nourhen Mezghanni, Nesrine Chaâri, Noureddine Ben Said.

**Project administration:** Mohamed Frikha.

**Resources:** Mohamed Frikha, Mohammed S. Alibrahim, Majed M. Alhumaid, Marwa M. Hassan, Mostafa S. Amira, Nasser Abouzeid.

**Software:** Mohamed Frikha, Noureddine Ben Said.

**Supervision:** Mohamed Frikha, Mohammed S. Alibrahim, Majed M. Alhumaid, Raghad S. Alharbi, Nasser Abouzeid.

**Validation:** Mohamed Frikha, Nesrine Chaâri, Noureddine Ben Said.

**Visualization:** Nourhen Mezghanni, Mohammed S. Alibrahim, Majed M. Alhumaid, Marwa M. Hassan, Raghad S. Alharbi, Mostafa S. Amira.

**Writing – original draft:** Mohamed Frikha, Nourhen Mezghanni, Nesrine Chaâri.

**Writing – review & editing:** Mohamed Frikha.

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
