## [Decision Letter · Decision Letter 0]

20 Nov 2023

PONE-D-23-28306Towards improving online physical education learning: gender differences and predictors associated to motivation, psychological needs satisfaction, and academic achievement in Saudi studentsPLOS ONE

Dear Dr. Frikha,

Thank you for submitting your manuscript to PLOS ONE. After careful consideration, we feel that it has merit but does not fully meet PLOS ONE’s publication criteria as it currently stands. Therefore, we invite you to submit a revised version of the manuscript that addresses the points raised during the review process.

We look forward to receiving your revised manuscript.

Kind regards,

Chao Gu

Academic Editor

PLOS ONE

Journal Requirements:

"The authors thank all of the subjects who participated in this study. Special thanks are given to the faculty members and IT staff administrations at KFU, KSU, and TU for their help, as well as to the Deanship of Scientific Research at King Faisal University for the financial support (GRANT 3961)."

"The corresponding author Frikha Mohamed received a grant for the present study.

GRANT 3961

Funder: The Deanship of Scientific Research, King Faisal University, Al-Ahsa 31982, Saudi Arabia

https://www.kfu.edu.sa/en/Deans/Research/Pages/Home-new.aspx

Reviewers' comments:

Reviewer's Responses to Questions

**Comments to the Author**

1. Is the manuscript technically sound, and do the data support the conclusions?

Reviewer #1: Yes

Reviewer #2: Yes

2. Has the statistical analysis been performed appropriately and rigorously? 

Reviewer #1: Yes

Reviewer #2: Yes

3. Have the authors made all data underlying the findings in their manuscript fully available?

Reviewer #1: Yes

Reviewer #2: Yes

4. Is the manuscript presented in an intelligible fashion and written in standard English?

Reviewer #1: Yes

Reviewer #2: Yes

5. Review Comments to the Author

Reviewer #1: I would like to thank for the opportunity to review this manuscript written on highly important topic. Overall, the manuscript is written in well quality, I have only some suggestions to improve the quality of this manuscript.

Title of the manuscript is informative and reads well. However, Authors could consider shortening the title of the manuscript.

Abstract is well written; I have now further comments.

Introduction is accurate and gives a great overview of the scientific background.

Methods is clear and written in a detail.

Authors provide sophisticated data analysis with very detailed results, great job!

The discussion is also very well written with a lot of comparisons to previous work. My only concern is that Authors could provide more suggestions for future research. Future research could rely on specific classification system of motivational behaviours. For example, in a recent study by Ahmadi et al. (2023), a list of motivational and behaviour change techniques were proposed for educational context which could be used in future intervention studies as a basis for intervention content.

Ahmadi, A., Noetel, M., Parker, P., Ryan, R. M., Ntoumanis, N., Reeve, J., Beauchamp, M., Dicke, T., Yeung, A., Ahmadi, M., Bartholomew, K., Chiu, T. K. F., Curran, T., Erturan, G., Flunger, B., Frederick, C., Froiland, J. M., González-Cutre, D., Haerens, L., . . . Lonsdale, C. (2023). A classification system for teachers’ motivational behaviors recommended in self-determination theory interventions. Journal of Educational Psychology. Advance online publication. https://doi.org/10.1037/edu0000783

Reviewer #2: The manuscript deals with an interesting and contemporary issue. However, the title "Towards online physical education learning:" is confusing; suggested to change the title to "Towards online learning in physical education:". The introduction needs some improvements: it should highlight the research gap with previous studies. Besides, it should status the objective of conducting the binary logistic regression.

The literature review and hypothesis sections are missing. The authors should provide definition of each variable, especially the PNS components in the literature review section, and explain its association with the research topic.

The materials and methods sections are adequate.

The results discussion needs to be further enhanced. It stressed an unequal samples t-test but didn't reveal whether equal or unequal variance was assumed. The discussion in section 3.3 didn't explain the parenthesis parameters (the meaning of small, medium and n square). In section 3.4, the authors elaborate on the comparison between group and gender without supporting results or tables. The "experience to OL" in Table 4 should clarified as "prior experience with OL". The exp (B) values reported in Table 6 were wrong. For positive beta, the exp (B) values should be greater than 1; for negative beta (AM), the exp (B) value should be less than 1.

Discussion: suggested the authors associate the findings with research objectives.

6. PLOS authors have the option to publish the peer review history of their article (what does this mean?). If published, this will include your full peer review and any attached files.

Reviewer #1: No

Reviewer #2: No

---

## [Author Response · Author response to Decision Letter 0]

28 Nov 2023

To Professor Chao Gu

The Academic Editor 11/26/2023

PLOS ONE

Dear Editor,

All co-authors of the manuscript (ID: PONE-D-23-28306) entitled “Towards improving online learning in physical education: gender differences and determinants of motivation, psychological needs satisfaction, and academic achievement in Saudi students” and I, want to thank you for the consideration to our work. We agree with the comments advanced by the reviewers and believe that the requested rectifications can improve the quality of the manuscript. 

Here you find the point-by-point comments and answers.

Answers point by point to the Editor and reviewers' comments

(Manuscript PONE-D-23-28306)

Answers to the Editor's questions:

Editor

 questions answers

1 1. Please ensure that your manuscript meets PLOS ONE's style requirements, including those for file naming. Style requirements verified. Changes in the headings and tables were made.

File naming verified.

2 We note that you have provided funding information that is not currently declared in your Funding Statement. However, funding information should not appear in the Acknowledgments section or other areas of your manuscript. We will only publish funding information present in the Funding Statement section of the online submission form. The funding information was removed from the acknowledgment section.

Please insert this statement in the online submission form:

Funding: This study was funded by the Deanship

of Scientific Research at King Faisal University,

Saudi Arabia (grant number GRANT3961). The funder had no role in study design, data collection and analysis, decision to publish, or preparation of the manuscript.

3 We note that you have stated that you will provide repository information for your data at acceptance. Should your manuscript be accepted for publication, we will hold it until you provide the relevant accession numbers or DOIs necessary to access your data. If you wish to make changes to your Data Availability statement, please describe these changes in your cover letter and we will update your Data Availability statement to reflect the information you provide. Data for the present are now published in Harvard Dataverse. Here is the DOI

https://doi.org/10.7910/DVN/WQWLMU

Answers to the reviewer 1 questions

Reviewer 1

 questions answers

1 The title of the manuscript is informative and reads well. However, Authors could consider shortening the title of the manuscript. To avoid any confusion in meaning, the title was reformed to "Towards improving online learning in physical education: gender differences and determinants of motivation, psychological needs satisfaction, and academic achievement in Saudi students". However, due to multiple study variables, its length was not reduced.

2 The authors could provide more suggestions for future research. Future research could rely on a specific classification system of motivational behaviors. A study perspective was added in relation to the findings of Ahmadi et al (2023).

Lines 738-742: Moreover, the teachers’ in-class behavior is considered a key factor that influences students’ motivation [58]. Thus, promoting students' motivation during PE online learning according to an individualized specific classification system of teachers' motivational behaviors remains an interesting future research topic too [58].

Answers to the reviewer 2 questions

Reviewer 2

 questions answers

1 the title "Towards online physical education learning:" is confusing To avoid any confusion in meaning, the title was reformed to "Towards improving online learning in physical education: gender differences and determinants of motivation, psychological needs satisfaction, and academic achievement in Saudi students".

2 The introduction needs some improvements: it should highlight the research gap with previous studies. Besides, it should status the objective of conducting the binary logistic regression. The introduction was improved. Indeed a paragraph related to the Cultural and social characteristics of the Saudi population was added. (L95-102). The introduction was divided into three sections.

Logistic regression is an extension of “regular” linear regression, used when the dependent variable, Y, is categorical (non-metric). The binary logistic regression, in studies in which the dependent variable is a “Yes/No” type variable. Typically we refer to the two categories of Y as “1” and “0,” so that they are represented numerically. This is the case of the GPA variable where students were divided into a high GPA group (3 < GPA ≤ 5); and a low GPA group (1 < GPA ≤ 3).

3 The literature review and hypothesis sections are missing. The authors should provide a definition of each variable, especially the PNS components in the literature review section, and explain its association with the research topic. The introduction was re-structured and the three study hypotheses were added at the end.

4 The results need to be further enhanced. It stressed an unequal samples t-test but did not reveal whether equal or unequal variance was assumed. Welch's t-test also known as the unequal variances t-test is used when you want to test whether the means of two populations are equal. This test is generally applied when there is a difference between the variations of two populations and also when their sample sizes are unequal, which corresponds to our study.

Thus in the present study, equal variance was not assumed in all comparisons.

Changes are made in the result section accordingly. 

5 The discussion in section 3.3 didn't explain the parenthesis parameters (the meaning of small, medium, and n square). Explanation was added in the data analysis section. Partial eta squared is a way to measure the effect size. According to Cohen (1992) it is classified as small (0.1), medium (0.3), and large (> 0.5) 

6 In section 3.4, the authors elaborate on the comparison between group and gender without supporting results or tables. The "experience to OL" in Table 4 should clarified as "prior experience with OL". The results presented (L314-350) are additional results to better understand the detected differences between variables and show the variation of PNS and motivation according to the GPA in males only and in females only. However, we do not opt to present a table in order to avoid repetitions. If the reviewer believes that is necessary we can add a table related to those results.

7 The exp (B) values reported in Table 6 were wrong. For positive beta, the exp (B) values should be greater than 1; for negative beta (AM), the exp (B) value should be less than 1. We fully agree with this remark. For negative B the Exp(B) should be less than 1 with is the case of amotivation variable. Indeed, by reviewing all the stat procedures and calculations, we discover mistakes in variable encoding generating errors in positive and negative beta values as well as in the Exp(B). However, the absolute beta values were correct. 

Results of the binary LR are fully presented in the result section (L385-469).

8 Discussion: suggested the authors associate the findings with research objectives. The discussion was revised and study limitations improved (L6955-698) (L739-742).

The corresponding author

Frikha Mohamed

---

## [Decision Letter · Decision Letter 1]

8 Jan 2024

PONE-D-23-28306R1Towards improving online learning in physical education: gender differences and determinants of motivation, psychological needs satisfaction, and academic achievement in Saudi studentsPLOS ONE

Dear Dr. Frikha,

Thank you for submitting your manuscript to PLOS ONE. After careful consideration, we feel that it has merit but does not fully meet PLOS ONE’s publication criteria as it currently stands. Therefore, we invite you to submit a revised version of the manuscript that addresses the points raised during the review process.

**ACADEMIC EDITOR: ** Please further explain in Tables 2, 3, and 4 what the 95% confidence intervals (95% CI) correspond to in terms of values. Moreover, in the abstract, please narrate in a storytelling manner without including specific numerical details related to statistics.

We look forward to receiving your revised manuscript.

Kind regards,

Chao Gu

Academic Editor

PLOS ONE

Journal Requirements:

Reviewers' comments:

Reviewer's Responses to Questions

**Comments to the Author**

1. If the authors have adequately addressed your comments raised in a previous round of review and you feel that this manuscript is now acceptable for publication, you may indicate that here to bypass the “Comments to the Author” section, enter your conflict of interest statement in the “Confidential to Editor” section, and submit your "Accept" recommendation.

Reviewer #1: All comments have been addressed

Reviewer #2: All comments have been addressed

2. Is the manuscript technically sound, and do the data support the conclusions?

Reviewer #1: Yes

Reviewer #2: Yes

3. Has the statistical analysis been performed appropriately and rigorously? 

Reviewer #1: Yes

Reviewer #2: Yes

4. Have the authors made all data underlying the findings in their manuscript fully available?

Reviewer #1: Yes

Reviewer #2: (No Response)

5. Is the manuscript presented in an intelligible fashion and written in standard English?

Reviewer #1: Yes

Reviewer #2: Yes

6. Review Comments to the Author

Reviewer #1: Authors have done well job on revising the manuscript. I have no further comments. Great job and thank you!

Reviewer #2: (No Response)

7. PLOS authors have the option to publish the peer review history of their article (what does this mean?). If published, this will include your full peer review and any attached files.

Reviewer #1: No

Reviewer #2: No

---

## [Author Response · Author response to Decision Letter 1]

11 Jan 2024

To Professor Chao Gu

The Academic Editor 10/01/2024

PLOS ONE

Dear Editor,

All co-authors of the manuscript (ID: PONE-D-23-28306) entitled “Towards improving online learning in physical education: gender differences and determinants of motivation, psychological needs satisfaction, and academic achievement in Saudi students” and I, want to thank you for the consideration to our work. Here you find the point-by-point answers to your comments.

Answers point by point to the Editor comments

(Manuscript PONE-D-23-28306)

(All changes in the manuscript are in red)

Editor

 questions answers

1 Please further explain in Tables 2, 3, and 4 what the 95% confidence intervals (95% CI) correspond to in terms of values. The 95% confidence interval defines a range of values that you can be 95% certain contains the population mean. In tables 2,3, and 4 the 95% confidence intervals correspond to the range of differences between means. Additional indications for CI were highlighted in the tables.

2 in the abstract, please narrate in a storytelling manner without including specific numerical details related to statistics. Abstract was revised and numerical details were removed.

 The corresponding author

---

## [Editor Report · Decision Letter 2]

15 Jan 2024

Towards improving online learning in physical education: gender differences and determinants of motivation, psychological needs satisfaction, and academic achievement in Saudi students

PONE-D-23-28306R2

Dear Dr. Frikha,

We’re pleased to inform you that your manuscript has been judged scientifically suitable for publication and will be formally accepted for publication once it meets all outstanding technical requirements.

Kind regards,

Chao Gu

Academic Editor

PLOS ONE

---

## [Editor Report · Acceptance letter]

28 Jan 2024

PONE-D-23-28306R2 

PLOS ONE

Dear Dr. Frikha, 

I'm pleased to inform you that your manuscript has been deemed suitable for publication in PLOS ONE. Congratulations! Your manuscript is now being handed over to our production team.

Kind regards, 

on behalf of

Dr. Chao Gu 

Academic Editor

PLOS ONE